# High Piezoelectric Performance of KNN-Based Ceramics over a Broad Temperature Range through Crystal Orientation and Multilayer Engineering

**DOI:** 10.3390/molecules29194601

**Published:** 2024-09-27

**Authors:** Guangrui Lu, Yunting Li, Rui Zhao, Yan Zhao, Jiaqi Zhao, Wangfeng Bai, Jiwei Zhai, Peng Li

**Affiliations:** 1School of Materials Science and Engineering, Liaocheng University, Liaocheng 252059, China; 2210180108@stu.lcu.edu.cn (G.L.); 2310180115@stu.lcu.edu.cn (Y.L.); 2021403640@sut.lcu.edu (R.Z.); 2021403623@stu.lcu.edu.cn (Y.Z.); 2021403612@stu.lcu.edu.cn (J.Z.); 2College of Materials and Environmental Engineering, Hangzhou Dianzi University, Hangzhou 310018, China; baiwangfeng@hdu.edu.cn; 3School of Materials Science and Engineering, Tongji University, Shanghai 201804, China; apzhai@tongji.edu.cn

**Keywords:** potassium sodium niobate, crystal texture, multilayer composite, piezoelectric performance, temperature stability

## Abstract

Uninterrupted breakthroughs in the room temperature piezoelectric properties of KNN-based piezoceramics have been witnessed over the past two decades; however, poor temperature stability presents a major challenge for KNN-based piezoelectric ceramics in their effort to replace their lead-based counterparts. Herein, to enhance temperature stability in KNN-based ceramics while preserving the high piezoelectric response, multilayer composite ceramics were fabricated using textured thick films with distinct polymorphic phase transition temperatures. The results demonstrated that the composite ceramics exhibited both outstanding piezoelectric performance (d_33_~467 ± 16 pC/N; S~0.17% at 40 kV/cm) and excellent temperature stability with d_33_ and strain variations of 9.1% and 2.9%, respectively, over a broad temperature range of 25–180 °C. This superior piezoelectric temperature stability is attributed to the inter-inhibitive piezoelectric fluctuations between the component layers, the diffused phase transition, and the stable phase structure with a rising temperature, as well as the permanent contribution of crystal orientation to piezoelectric performance over the studied temperature range. This novel strategy, which addresses the piezoelectric and strain temperature sensitivity while maintaining high performance, is well-positioned to advance the commercial application of KNN-based lead-free piezoelectric ceramics.

## 1. Introduction

Piezoelectric materials enable the conversion of mechanical energy into electrical energy and vice versa. Piezoelectric materials are an essential component in various technological applications due to their unique electromechanical conversion properties. Notably, piezoelectric ceramics are extensively utilized in areas such as sensors, actuators, aerospace engineering, medical imaging systems, and precision control systems, etc. [1,2,3,4,5]. Despite their widespread use, most piezoelectric ceramics in current use contain lead, a toxic substance harmful to both the environment and human health [6,7]. This has led to increasing concern and an urgent need to develop lead-free alternatives. Among the potential candidates, potassium sodium niobate [(K,Na)NbO_3_, KNN] piezoelectric ceramics are distinguished by their relatively superior electromechanical properties and higher Curie temperature (T_c_ = 435 °C) compared to other lead-free alternatives [5,8,9,10,11,12,13].

Although KNN piezoelectric ceramics are environmentally friendly alternatives to lead-based materials, they also face challenges, including a lower piezoelectric coefficient (d_33_ < 80 pC/N) and inferior temperature stability related to the polycrystalline phase boundary [14]. From the previously reported research, several effective strategies have been proposed to enhance the electromechanical properties of KNN-based ceramics. The main approaches include phase boundary engineering, domain engineering, and the development of single-crystal and textured ceramics. Phase boundary engineering focuses on optimizing material composition by employing element doping or combining KNN with other components to raise the rhombohedral–orthorhombic phase transition temperature (T_R-O_) or lower the orthorhombic–tetragonal phase transition temperature (T_O-T_), thereby constructing a two-phase or multiphase coexisting structure at room temperature. By virtue of the reduced polarization anisotropy caused by the phase boundary, KNN-based piezoelectric ceramics can achieve enhanced piezoelectric activity [15,16]. For instance, Wang et al. developed KNN-based lead-free piezoelectric ceramics with O-T phase coexistence at room temperature by partially substituting K with Li and Nb with Ta and Sb, which resulted in a piezoelectric coefficient (d_33_) of up to 308 pC/N and excellent electromechanical coupling properties (k_p_~51%, k_t_~47%) [17]. Domain engineering aims to maximize the piezoelectric coefficient by manipulating the size of the electric domains, as well as the movement and orientation of the domain walls. Furthermore, the texturing process, which induces grain orientation growth by adding templates, can significantly enhance the piezoelectric coefficient without lowering the Curie temperature. For example, in 2004, Saito et al. reported KNN-based textured ceramics with O-T phase coexistence at room temperature, achieving a piezoelectric coefficient (d_33_) of 416 pC/N, while in 2018, Li et al. developed <001>-textured KNN-based ceramics with R-O phase coexistence, exhibiting a d_33_ value as high as 700 pC/N [18,19].

Despite these advances, significant challenges persist in achieving a balance between high piezoelectric performance and temperature stability for KNN-based piezoceramics. These strategies that have been reported to improve the temperature stability of piezoelectric performance involve the creation of diffused phase boundaries and the design of compositionally graded structures [20,21]. For instance, Li et al. developed KNN-based piezoelectric ceramics that exhibited good temperature stability by constructing diffused polymorphic phase boundaries and achieved a d_33_ variation rate of less than 10% in the range of room temperature to 80 °C [22]. Another approach is the gradual variation in material composition, known as compositional grading, which involves combining a series of KNN-based piezoelectric components with different phase transition temperatures. This method utilizes the complementary effect of composite ceramics to maintain a stable phase structure within a certain temperature range, thereby mitigating the impact of phase structure changes on the temperature stability of piezoelectric performance. For example, Li et al. developed KNN-based composite ceramics characterized by consecutive phase transition gradients via layered distribution of dopants, resulting in both high piezoelectric performance (d_33_ = 508 pC/N at ambient temperature) and excellent temperature stability (a variation in d_33_ of less than 13%, and the unipolar strain (S_uni_) remained within 8% over the temperature range of 25–150 °C) [23].

In order to develop KNN-based piezoelectric ceramics that combine high piezoelectric performance with excellent temperature stability, herein, a novel approach was proposed to fabricate multilayer composites using textured thick films. The research builds upon prior findings by selecting specific compositions with distinct polymorphic phase transition temperatures: 0.97 (K_0.5_Na_0.5_) (Nb_0.96_Sb_0.04_) O_3_-0.02BaZrO_3_-0.01 (Bi_0.5_K_0.5_) HfO_3_ (denoted as KNN-T_1_) and 0.95 (K_0.5_Na_0.5_) (Nb_0.96_Sb_0.04_)O_3_-0.02BaZrO_3_-0.03 (Bi_0.5_K_0.5_) HfO_3_ (denoted as KNN-T_3_). The O-T phase transition temperature is far from room temperature for KNN-T_1_, while the R-O-T phase transition temperature is at approximately room temperature. Therefore, the d_33_ and S_uni_ values increase significantly with the increase in temperature for KNN-T_1_, while they decrease gradually with elevated temperature for KNN-T_3_. In view of the distinct structural and performance features, KNN-T_1_ and KNN-T_3_ were selected to synthesize the multilayer composite ceramics through the stacking of textured thick films in a specific ratio. The results indicate that the multilayer composites made from KNN-T_1_ and KNN-T_3_ exhibit enhanced piezoelectric properties and exceptional temperature stability, offering a promising solution for high-performance, lead-free piezoelectric materials. This study not only advances the development of KNN-based ceramics but also paves the way for future research in developing environmentally friendly piezoelectric materials.

## 2. Results and Discussion

Figure 1a shows the schematic diagram for preparing multilayer composite ceramics by stacking textured thick films KNN-T_1_ and KNN-T_3_ in accordance with a ratio of 1:6, which is determined to be the optimal composite ratio by comparing both piezoelectric properties and temperature stability among all designs. To obtain the phase structure information of each component layer in the composite ceramic, a slant surface (red area) was cut out (Figure 1b), and XRD tests were conducted, resulting in the XRD patterns of the KNN-T_1_ and KNN-T_3_ layers, as well as multilayer composite textured ceramics, as shown in Figure 1c. All samples exhibited a typical perovskite structure without the presence of secondary phases, indicating a uniform solid solution was formed in the multilayer composite ceramics. Notably, the relative intensities of the (001)/(100) and (002)/(200) diffraction peaks were much higher than other diffraction peaks in all samples. The Lotgering orientation factors F_001_ were determined to be 83%, 89%, and 88% for KNN-T_1_, KNN-T_3_, and 1:6 multilayer composite ceramics, respectively, indicating a good alignment of the grains along the <001> direction.

Figure 2a displays the temperature-dependent dielectric constant (ɛ_r_-T) curves for KNN-T_1_, KNN-T_3_, and 1:6 multilayer composite textured ceramics. The observations reveal that: (i) KNN-T_1_ ceramics exhibit three distinct dielectric anomaly peaks away from room temperature, corresponding to rhombohedral–orthorhombic (R-O) phase transition, orthorhombic–tetragonal (O-T) phase transition, and tetragonal–cubic (T-C) phase transition; and (ii) KNN-T_3_ ceramics and 1:6 multilayer composite ceramics have only two dielectric anomaly peaks. This occurs because, with increasing (Bi_0.5_K_0.5_) HfO_3_ (BKH) concentration, the R-O and O-T phase transition temperatures gradually approach room temperature and merge to form a rhombohedral–orthorhombic–tetragonal (R-O-T) phase transition, as marked in Figure 2a, which is consistent with previous studies [24,25,26,27]. The high temperature dielectric anomaly is attributed to the tetragonal–cubic (T-C) phase transition. The room temperature phase structure of the samples was evaluated by the intensity ratio of the (002) and (200) peaks (Figure 1d) and the ɛ_r_-T curves (Figure 2a). For the sample with a single orthorhombic phase, the intensity ratio of the (002) and (200) peaks (I_(002)_/I_(200)_) is about 2:1, while for the sample with single tetragonal phase, it is approximately 1:2 [28,29]. Thus, it is inferred that the phase structure of KNN-T_1_ ceramics is primarily the orthorhombic (O) phase, whereas KNN-T_3_ and 1:6 multilayer composite ceramics possess a phase structure with coexisting R-O-T phases (i.e., polymorphic phase boundary, PPB). Additionally, KNN-T_3_ and 1:6 multilayer composite ceramics show broadening of the dielectric peak for the R-O-T phase transition, which indicates enhanced relaxor behavior. The degree of diffusion can be evaluated from the ɛ_r_-T curves by fitting the modified Curie–Weiss law [30]:$$\frac{1}{\varepsilon }-\frac{1}{\varepsilon_{m}}=\frac{{(T-T_{m})}^{\gamma }}{C}$$
where ε is the dielectric constant, ε_m_ is the maximum dielectric constant, T is temperature, T_m_ is the temperature at which the dielectric constant reaches its maximum, γ is the degree of diffusion, and C is the Curie constant. The value of γ ranges from 1 to 2, where higher values indicate stronger relaxor behavior. As shown in Figure 2b, the γ value of 1:6 multilayer composite ceramics is as high as 1.82, while those of KNN-T_1_ and KNN-T_3_ are 1.65 and 1.79, respectively, confirming a more diffused phase transition in the 1:6 multilayer composite ceramics. Figure 3 presents cross-sectional SEM images of all samples. It is evident that all samples have a dense microstructure and the grains’ growth is perpendicular to the tape casting direction, i.e., crystallography <001> direction. In addition, no specific grain morphology was observed in the cross-section of the multilayer composite ceramics, which can be attributed to two aspects. Firstly, the liquid phase formed during the high temperature sintering (1180 °C) blurred the grain morphology. Secondly, the SEM morphology was captured from the cross-section of the cutting-off sample, where trans-granular fracturing occurred.

Figure 4a shows the quasi-static piezoelectric coefficient d_33_ at room temperature for KNN-T_1_, KNN-T_3_, and 1:6 multilayer composite ceramics. The 1:6 multilayer composite ceramics exhibit a d_33_ value of up to 467 ± 16 pC/N, much higher than the d_33_ value of KNN-T_1_ ceramics (253 ± 15 pC/N) and slightly lower than the d_33_ value of KNN-T_3_ ceramics (484 ± 18 pC/N). The high piezoelectric response of 1:6 ceramics can be attributed to the coexistence of R-O-T phases as well as the crystal texture. The polymorphic phase boundary of the R-O-T multiphase structure reduces the energy barrier for polarization rotation, facilitating saturation polarization under an external electric field, thereby enhancing the piezoelectric response. The piezoelectric anisotropy resulting from the grain orientation along the <001> direction, and the advantageous “4R” and “4O” engineered domain configurations formed by the polarization of the rhombohedral and orthorhombic phases along the <001> direction, contribute to the high piezoelectric properties [21,31,32,33,34]. As compared to the monomer KNN-T_3_ ceramics, the slight reduction in d_33_ for the 1:6 multilayer composite ceramics can be attributed to the effect of the component layer KNN-T_1_ with low piezoresponse on the overall piezoelectric properties of the composite ceramics. In addition, the variation in R/O/T phase content and the interface between the constituent layer may also deteriorate the piezoelectric properties to some extent for the 1:6 multilayer composite ceramics [35]. From the practical application point of view, the temperature stability of piezoelectric properties is crucial. Figure 4b shows the temperature-dependent P-E loops of 1:6 multilayer composite ceramics measured at a frequency of 10 Hz and an electric field of 40 kV/cm. All P-E loops maintain a saturated hysteretic characteristic, while a slight decrease in remnant polarization (P_r_) with elevating temperature was observed. To evaluate the temperature stability of d_33_, in situ measurements of d_33_ (E) hysteresis loops at varying temperatures were conducted, and the results are shown in Figure 4c. The small signal piezoelectric coefficient d_33_ was extracted approximately from the d_33_ (E) loops at E = 0, and the data were normalized, as illustrated in Figure 4d [30,36]. One can see the d_33_ values of KNN-T_1_ ceramics first significantly increase in the temperature range of 25 °C-T_O-T_, and then rapidly decline when the temperature exceeds T_O-T_, exhibiting a variation ratio as high as 66.2% relative to their room temperature values over the same temperature range. The d_33_ values of KNN-T_3_ ceramics almost monotonically decrease from 25 to 180 °C, likewise showing a large variation of 64.6%. As compared to each constituent layer (KNN-T_1_ and KNN-T_3_), the d_33_ values of 1:6 multilayer composite ceramics exhibit a slight fluctuation with elevating temperature, giving rise to a variation ratio as low as 9.1% over a wide temperature range from 25 to 180 °C, indicating a marked enhancement in the temperature stability of d_33_. Compared to single-component ceramics, the d_33_ temperature stability of 1:6 multilayer composite ceramics is markedly enhanced. It is worth noting that the d_33_ can still maintain up to 426 pC/N when the temperature is as high as 180 °C. Figure 4e compares the temperature stability of d_33_ (denoted as ∆d_33_, ∆d_33_ = (d_33,T_-d_33,RT_)/d_33,RT_ × 100%) between the 1:6 multilayer composite ceramics and some previously reported KNN-based piezoelectric ceramics [23,37,38,39,40,41,42]. It can be clearly observed that previously reported KNN-based ceramics always exhibit large fluctuations (>10%) of d_33_ values with increasing temperature, while the 1:6 multilayer composite ceramics reported in this work only show a variation below 10% over a broad temperature range (25–180 °C). The excellent temperature stability of the 1:6 multilayer composite ceramics may benefit from the diffused phase transition and the complementary effects of each constituent layer with a distinct phase transition temperature [22,37]. According to the phenomenological relationship d_33_ = 2Q⋅ε_r_⋅P_r_, d_33_ is strongly correlated with the remnant polarization P_r_ and the relative dielectric constant ε_r_ [43]. Thus, the P_r_⋅ε_r_ values with rising temperature relative to their room temperature values (P_r_⋅ε_r_ (T)/P_r_⋅ε_r_ (RT)) for KNN-T_1_, KNN-T_3_, and 1:6 multilayer composite ceramics are given in Figure 4f. It is evident that compared to KNN-T_1_ and KNN-T_3_ ceramics, the P_r_⋅ε_r_ of 1:6 multilayer composite ceramics demonstrates superior temperature stability, which contributed to the excellent temperature stability of the d_33_ of 1:6 ceramics.

In order to reveal the effect of phase structure with temperature on the piezoelectric temperature stability, the in situ temperature-dependent XRD measurement was performed, and the results are presented in Figure 5a. The representative results of the Rietveld refinement of XRD patterns measured at 25 °C, 85 °C, 150 °C, and 180 °C are given in Figure 5b–e. The low values of the weighted pattern reliability factor (R_wp_ ≤ 3.87%) and pattern reliability factor (R_p_ ≤ 3.00%), and the goodness of fit indicator (χ^2^ ≤ 1.59) indicate that all XRD patterns are accurately refined and the selected phase structures are reliable. The structural refinement results show that the 1:6 multilayer composite ceramics exhibit coexisting R, O, and T phases within the temperature range of 25 °C to 180 °C. Moreover, as the temperature increases, the content of the R and O phases gradually decreases, while the content of the T phase increases, which is consistent with the temperature-dependent dielectric constant curves (Figure 2a). Therefore, we can deduce that the stable phase structure with rising temperature also contributed to the excellent temperature stability of piezoelectric properties.

In addition to piezoelectric temperature stability, the strain temperature dependence is also an important indicator for piezoelectric actuator application. The unipolar electric-field-induced strain (S_uni_) of approximately 0.17% at 40 kV/cm and room temperature was observed in the 1:6 multilayer composite ceramics, which is between that of constituent layers KNN-T_1_ (S_uni_~0.12%) and KNN-T_3_ (S_uni_~0.18%), as shown in Figure 6a. To investigate the strain temperature stability, the in situ temperature-dependent strain test was conducted in the temperature range of room temperature to 180 °C, as shown in Figure 6b. The temperature stability of the strain for KNN-T_1_, KNN-T_3_, and 1:6 multilayer composite ceramics was compared in Figure 6c using the calculated values of S_uni_ (T)/S_uni_ (RT), which represent the ratio of electric-field-induced strain measured at a specific temperature (T) to that measured at room temperature (RT). The results indicate that the 1:6 ceramics exhibit good temperature stability of strain, with a variation rate of less than 2.9% over a broad temperature range of room temperature to 180 °C, which is significantly lower than that of the constituent layers of KNN-T_1_ (~57.4%) and KNN-T_3_ (~10.6%). Notably, the strain temperature stability of the 1:6 ceramics is far superior to that of previously reported KNN-based lead-free ceramics, and even better than some lead-based piezoelectric ceramics, as displayed in Figure 6d [41,43,44,45,46,47]. The excellent strain temperature stability of the 1:6 multilayer composite ceramics can be attributed to their strong dispersion characteristic of phase transition, and the stable phase structure with respect to temperature from room temperature to 180 °C. The above results demonstrate that the 1:6 multilayer composite ceramics exhibit promising development prospects in high-temperature transducer and actuator applications.

## 3. Materials and Methods

### 3.1. Materials and Synthesis

Textured thick films 0.97(K_0.5_Na_0.5_)(Nb_0.96_Sb_0.04_)O_3_-0.02BaZrO_3_-0.01(Bi_0.5_K_0.5_)HfO_3_ (abbreviated as KNN-T_1_) and 0.95(K_0.5_Na_0.5_)(Nb_0.96_Sb_0.04_)O_3_-0.02BaZrO_3_-0.03(Bi_0.5_K_0.5_)HfO_3_ (abbreviated as KNN-T_3_) were fabricated via the template grain growth (TGG) method. Initially, the KNN-T_1_ and KNN-T_3_ powders were synthesized through the conventional high-temperature solid-state reaction method. High-purity reagents, K_2_CO_3_ (99%), Na_2_CO_3_ (99.95%), Nb_2_O_5_ (99.98%), Sb_2_O_3_ (99.9%), BaCO_3_ (99.99%), ZrO_2_ (99%), Bi_2_O_3_ (99.975%), and HfO_2_ (99.99%) powders were weighed according to stoichiometric ratios and mixed in anhydrous ethanol by ball milling for 24 h, then calcined at 850 °C for 5 h to form the primary perovskite structure. The calcined powders were subjected to an additional 48 h of ball milling to ensure a sufficiently fine particle size. Subsequently, to obtain film slurries, the matrix powders were mixed with 3 mol% NaNbO_3_ templates, 0.5 wt.% MnO_2_ sintering aid, 40 wt.% polyvinyl butyral (PVB) binder, and 170 wt.% solvents (a mixture of anhydrous ethanol and methylbenzene in a 2:1 weight ratio) using a roller mill mixer. The uniform slurry was then tape-cast into green tapes with a thickness of 30–60 μm using a doctor blade. After drying, the KNN-T_1_ and KNN-T_3_ green tapes were cut, stacked in ratios of KNN-T_1_:KNN-T_3_ = 1:x (x = 3, 4, 5, 6), followed by uniaxial pressing at 12 MPa and 65 °C for 5 min. After removing the organic binder at 600 °C, the samples were sintered using a two-step process, i.e., they were first heated to 1180 °C at a rate of 3 °C/min without dwell time, and then rapidly cooled to 1080 °C at a rate of 10 °C/min and held for 10 h. It is worth noting that in order to improve the orientation of grains and density of the textured ceramics, a relative high temperature of 1180 °C (higher than the melting temperature of KNN) was specially designed in the first step to form some liquid phase on the surface of NaNbO_3_ templates.

### 3.2. Characterization

The room temperature phase structure and grain orientation, as well as temperature-dependent phase evolution, were characterized by X-ray diffractometer (XRD, TD-3700, Dandong Tongda Technology Co., Ltd., Dandong, China). Full-profile Rietveld refinements of XRD data were performed using the EXPGUI 3.0 software package. The microstructure of the sintered samples was examined using field-emission scanning electron microscopy (FE-SEM, Carl Zeiss, Oberkochen, Germany). The temperature dependence of dielectric behavior was measured using an LCR meter (E4980A, Agilent Technologies, Santa Clara, CA, USA) connecting a computer-controlled heating and cooling system. The polarization hysteresis (P-E) loops, electric-field-induced strain (S-E) curves, and d_33_ (E) hysteresis loops were assessed using a ferroelectric analysis system (TF Analyzer 2000, aixACCT, Frankfurt, Germany) coupled with a laser interferometric vibrometer (AE SP-S120 E, SIOS Meßtechnik GmbH, Lower Saxony, Germany) and a temperature controller (TFA 401-8, aixACCT, Frankfurt, Germany). The d_33_ values were determined using a quasi-static d_33_ m (ZG-4, Institute of Acoustics, Beijing, China) after the samples were poled at 20 kV/cm for 20 min.

## 4. Conclusions

In summary, the novel strategy of creating a multilayer composite using textured thick films with distinct phase transition behavior effectively mitigates the piezoelectric and strain temperature sensitivity of KNN-based piezoelectric ceramics. A high piezoelectric response (d_33_~467 ± 16 pC/N) and unipolar strain (S_uni_~0.17%) were achieved in the 1:6 multilayer composite ceramics. Benefiting from the complementary effects of piezoelectric fluctuations between constituent layers of KNN-T_1_ and KNN-T_3_, a diffused phase transition, and a stable phase structure with respect to temperature as well as the grain orientation, the 1:6 multilayer composite ceramics exhibit excellent piezoelectric and strain temperature stability with variations in small signal d_33_ and strain of less than 9.1% and 2.9%, respectively, over a broad temperature range from room temperature to 180 °C. The strategy proposed in this work successfully addresses the critical challenge of temperature-sensitive piezoelectric performance for KNN-based lead-free piezoelectric ceramics, further promoting their commercial applications in high-temperature transducers and actuators.

## Figures and Tables

**Figure 1 molecules-29-04601-f001:**
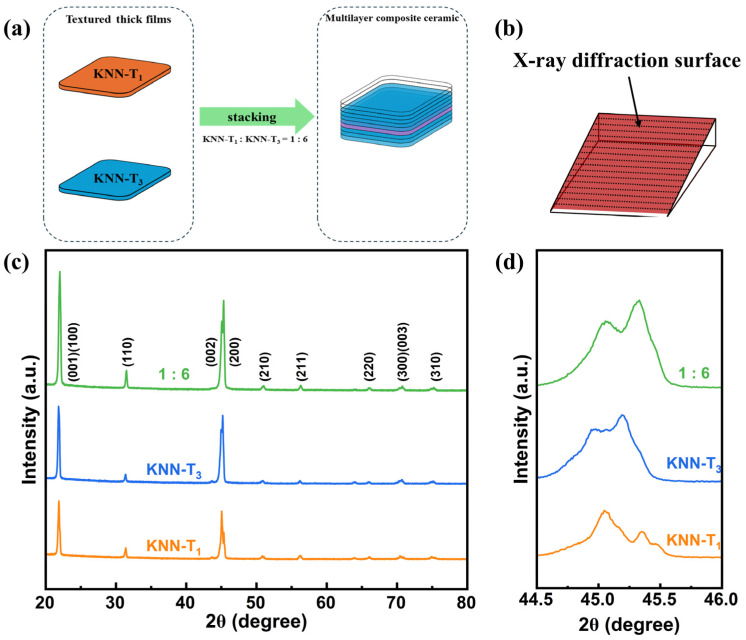
(**a**) Schematic diagram for preparing multilayer composite ceramics by stacking textured thick films KNN-T_1_ and KNN-T_3_ in accordance with a ratio of 1:6. (**b**) Schematic diagram of X-ray diffraction surface. XRD patterns of KNN-T_1_, KNN-T_3_, and 1:6 multilayer composite ceramics in the 2θ ranges of 20–80° (**c**), and 44.5–46° (**d**) measured at room temperature.

**Figure 2 molecules-29-04601-f002:**
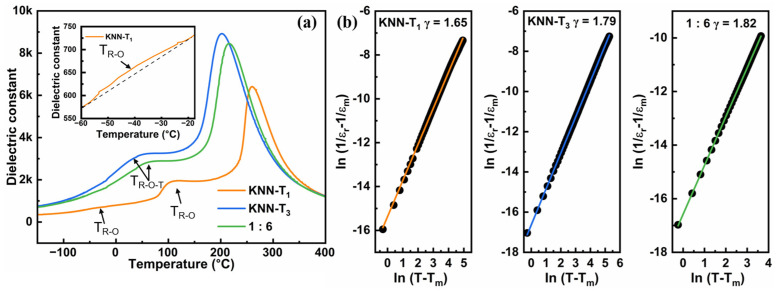
(**a**) Temperature-dependent dielectric constant curves measured at 100 kHz. The inset in the top left of (**a**) is the enlarged ε_r_ − T curves in the temperature range of −60 °C~−20 °C. (**b**) The degree of diffuseness γ calculated by fitting ln (1/ε_r_ − 1/ε_m_) versus ln (T-T_m_) curves.

**Figure 3 molecules-29-04601-f003:**
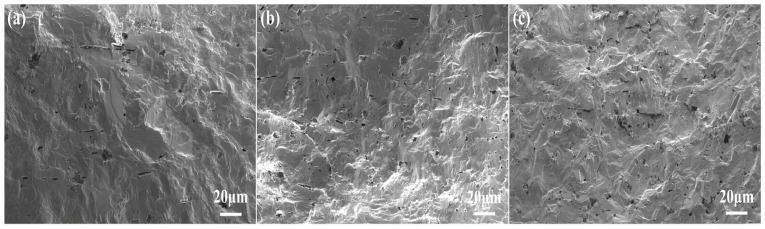
Cross-sectional SEM images of (**a**) KNN-T_1_; (**b**) KNN-T_3_; (**c**) 1:6 multilayer composite ceramics.

**Figure 4 molecules-29-04601-f004:**
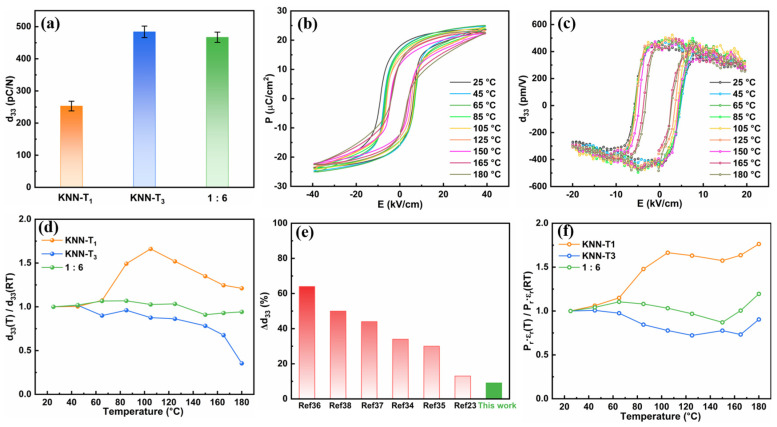
(**a**) d_33_ values of KNN-T_1_, KNN-T_3_, and 1:6 multilayer composite textured ceramics at room temperature. Temperature dependence of P-E (**b**) and d_33_ (E) loops (**c**) of 1:6 multilayer composite ceramics measured in the temperature range of 25 °C to 180 °C. (**d**) d_33_ (T)/d_33_ (RT) of KNN-T_1_, KNN-T_3_ and 1:6 ceramics as a function of temperature. (**e**) Comparison of piezoelectric temperature stability (denoted as Δd_33_) between the 1:6 multilayer composite ceramics and other piezoelectric ceramics reported in the literature. (**f**) Temperature-dependent P_r_⋅ε_r_ (T)/P_r_⋅ε_r_ (RT) of KNN-T_1_, KNN-T_3_ and 1:6 ceramics.

**Figure 5 molecules-29-04601-f005:**
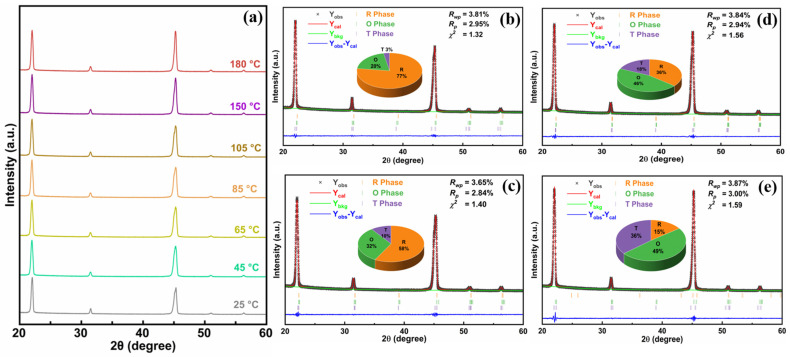
(**a**) XRD patterns of 1:6 multilayer composite ceramics measured at various temperatures. Rietveld refinement for the representative XRD patterns obtained at the temperatures of 25 °C (**b**), 85 °C (**c**), 150 °C (**d**), and 180 °C (**e**).

**Figure 6 molecules-29-04601-f006:**
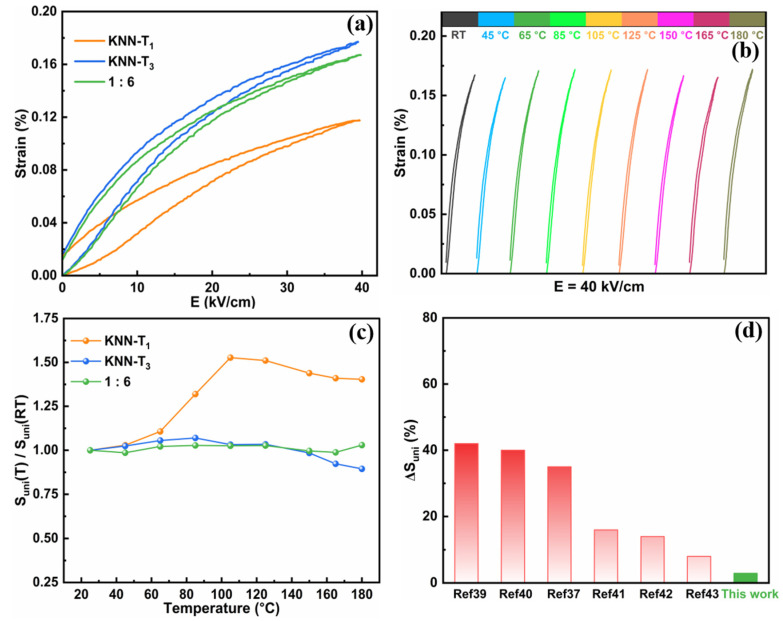
(**a**) Unipolar electric-field-induced strain for KNN-T_1_, KNN-T_3_, and 1:6 multilayer composite ceramics measured at room temperature. (**b**) Temperature-dependent unipolar strain curves of 1:6 ceramics measured in the temperature range of room temperature to 180 °C. (**c**) S_uni_ (T)/S_uni_ (RT) of KNN-T_1_, KNN-T_3_, and 1:6 ceramics as a function of temperature. (**d**) Comparison of temperature stability of strain (denoted as ΔS_uni_) between the 1:6 multilayer composite ceramics and other piezoelectric ceramics reported in the literature.

## Data Availability

Data are contained within the article.

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
