# Peer review of "High Piezoelectric Performance of KNN-Based Ceramics over a Broad Temperature Range through Crystal Orientation and Multilayer Engineering"

_molecules, 2024, doi:10.3390/molecules29194601_

Round 1
Reviewer 1 Report
Comments and Suggestions for Authors
This research article investigates the structural and electrical properties of KNN-based textured multilayer composites after sintering. Despite achieving very good results regarding the electrical properties, I have serious concerns about the process conducted.
It is a well-known issue that the melting temperature of KNN is at around 1140-1150°C (https://doi.org/10.1016/j.oceram.2024.100541 and https://doi.org/10.1016/j.jeurceramsoc.2024.116701). What is the reason behind selecting 1180°C in the two-step sintering process? What was the heating rate used?
KNN-based materials typically exhibit a specific grain morphology on the fracture surface. However, this morphology is not observed in Figure 3. Could you please explain the reason for this?
On page 4, line 126, where is the reference for the statement made? A conclusion on this trend cannot be drawn with only two compositions.
What was the frequency used for the polarization measurements? The authors should mention about the frequency value of the analysis.
The authors should explain the strain increase (Suni(T)/Suni(RT)) observed in KNN-T1 in Figure 6(c).
On Page 8, line 281, “… 170 wt.% solvents (a mixture of anhydrous ethanol and methylbenzene in a 2:1 weight ratio)….” Is the percentage correct?
Reviewer 2 Report
Comments and Suggestions for Authors
This study investigated crystal structure, microstructure, dielectric properties and electrical properties of KNN-based the textured composites ceramics. In fact, concept and purpose of this study seem to be good enough. However, the design of experiment and explanations (arrangement) of results are not suitable for accepting in Molerculers. Therefore, it is suggested that major revision are needed before acceptance. Detailed comments are shown here.
1. In Introduction part (in fact, this comment should be considered revision for whole parts of this paper), the aim and purpose of this study was well introduced by authors. However, it seems that explanation part for reason of the selected two materials is relatively poor. In other words, the main concept of this study is pretty same with previous study [ref. 23] expect for texturing. Therefore, it strongly recommends that authors should explain and discuss why two materials (KNN-T1 and KNN-T3) were selected to emphasize the novelty and creativity of this study in Introduction part (as introducing materials) and Results and discussion part (as expressing distinct features for two materials).
2. In Figure 1, as it mentioned earlier, authors should provide distinct features for each samples to explain structural features. In fact, only texturing can not be guarantee on novelty and creativity of this study. Therefore, please explain and discuss each of materials. Additionally, is there any reason why the ratio for stacked multilayer composite ceramics should be 1:6 (KNN-T1 : KNN-T3).
3. Authors claimed that "(i) KNN-T1 ceramics exhibit three distinct dielectric anomaly peaks away from 122 room temperature, corresponding to rhombohedral-orthorhombic (R-O) phase transition..". However, distinct evidence of R-O phase transition was not detected in Figure 2. Please make sure this part. What is meaning of "BKH" in 1st paragraph of page 4? Furthermore, authors claimed "1:6 134 multilayer composite ceramics possess a phase structure with coexisting R-O-T phases 135 (i.e., polymorphic phase boundary, PPB).". However, the provided result is insufficient for supporting author's explanation.
4. It seems that the provided results for temperature stability (both of d33 and strain properties) of 1:6 multilayer composite textured ceramics are difficult to find the advantage in comparison with a single KNN-T3.
Comments on the Quality of English LanguageN/A
Author Response
|
Thank you very much for taking the time to review this manuscript entitled “High piezoelectric performance of KNN-based ceramics over a broad temperature range through crystal orientation and multilayer engineering” (Manuscript ID: molecules-3181736). Those comments are all valuable and very helpful for revising and improving our paper, as well as the important guiding significance to our researches. We have studied comments carefully and have made correction which we hope meet with approval. Please find the detailed responses below and the corresponding revisions/corrections in the re-submitted files. We would like to express our great appreciation to you for comments on our paper. |
Looking forward to hearing from you.
Thank you and best regards.
Yours sincerely,
Peng Li
Comments 1: In Introduction part (in fact, this comment should be considered revision for whole parts of this paper), the aim and purpose of this study was well introduced by authors. However, it seems that explanation part for reason of the selected two materials is relatively poor. In other words, the main concept of this study is pretty same with previous study [ref. 23] expect for texturing. Therefore, it strongly recommends that authors should explain and discuss why two materials (KNN-T1 and KNN-T3) were selected to emphasize the novelty and creativity of this study in Introduction part (as introducing materials) and Results and discussion part (as expressing distinct features for two materials).
Response: Thank you very much for your constructive comments and suggestions. These comments are all valuable and very helpful for revising and improving our paper, as well as the important guiding significance to our researches. Up to now, it has been a bottleneck to enhance the piezoelectric properties and their temperature stability simultaneously for KNN-based lead-free ceramics only by means of chemical composition or phase boundary design. In order to realize high piezoelectric properties and excellent temperature stability, in this work, a combined strategy of chemical composition modification, multilayer composite, and texturing engineering was proposed to overcome the limitations affecting temperature stability. In this work, the reasons for selecting KNN-T1 and KNN-T3 to synthesis multilayer composite ceramics can be summarized into two aspects. Firstly, the piezoelectric coefficient (d33) and electric field-induced strain (Suni) decrease gradually for single KNN-T3 ceramics, while the d33 and Suni values increase first and then decrease for single KNN-T1 ceramics with elevated temperature from 25 oC to 180 oC, as shown in Figure 1(a). Therefore, in order to realize the complementary effect of variable piezoelectric properties with temperature, the KNN-T1 and KNN-T3 compositions were selected to synthesize multilayer composite ceramics. Secondly, the composite of KNN-T1 and KNN-T3 in a certain proportion (1:6) can enhance the relaxor behavior, and thus improve the temperature stability of piezoelectric properties. The interpretation and discuss why two materials (KNN-T1 and KNN-T3) were selected in this study have been added to the revised manuscript. (Page 2, line 91-97)
It is really true as Reviewer stress that the main concept of this study is slightly same with previous study [ref. 23] expect for texturing. It is worth noting that it is still difficult to substantially enhance the temperature stability of piezoelectric properties for KNN-based ceramics only via multilayer composite of two or more components. Our previous study suggested that crystallographic texturing is critical for improving piezoelectric properties and their temperature stability. Therefore, a combined strategy of chemical composition modification, multilayer composite, and texturing engineering is the novelty and creativity of this study. As a result, high piezoelectric properties and excellent temperature stability over a broad temperature range of 25 oC to 180 oC were successfully achieved by means of the combination strategy in the 1:6 multilayer composite textured ceramics.

Figure 1. (a) d33(T)/d33(RT) and (b) Suni(T)/Suni(RT) of KNN-T1 and KNN-T3 single ceramics as well as 1:6 multilayer composite ceramics as a function of temperature.
Comments 2: In Figure 1, as it mentioned earlier, authors should provide distinct features for each samples to explain structural features. In fact, only texturing can not be guarantee on novelty and creativity of this study. Therefore, please explain and discuss each of materials. Additionally, is there any reason why the ratio for stacked multilayer composite ceramics should be 1:6 (KNN-T1 : KNN-T3)
Response: Thank you very much for your constructive and valuable comments. The structural features of each sample were evaluated by means of XRD patterns and temperature-dependent dielectric constant (εr-T) curves. The phase structure of KNN-based ceramics can be quantified by assessing relative intensities of (002) and (200) peaks. For the orthorhombic phase, the ratio (I002/I200) is about 2:1, while transformed into 1:2 for the tetragonal phase. As for the KNN-T1 ceramic, the intensity ratio of the (002) and (200) peaks (I002/I200) is approximately 2:1. Furthermore, it can be seen from the εr-T curves the R-O and O-T phase transition temperatures are all far from room temperature. Therefore, we can infer that KNN-T1 ceramic has an orthorhombic phase structure at room temperature. However, as for the KNN-T3 and 1:6 multilayer composite ceramics, the intensity ratios of the (002) and (200) peaks are between 2:1 and 1:2. Moreover, the R-O and O-T phase transitions were merged into a single phase transition. Therefore, the phase structures of KNN-T3 and 1:6 multilayer composite ceramics were determined to be the coexistence of R, O and T phases. The results of Rietveld refinement of XRD patterns further prove the phase structure. The piezoelectric coefficient (d33) and strain (Suni) increase first and then decrease with the increase of temperature, reaching maximum values at the temperature approximately TO-T for KNN-T1 ceramics, while the d33 and Suni values decrease monotonously with elevated temperature from room temperature to 180 oC. In view of such structural and performance features, KNN-T1 and KNN-T3 materials were selected to synthesize multilayer composite ceramics to realize excellent temperature stability of piezoelectric properties by taking advantage of mutual compensatory effects. The combinatorial optimization of phase structure and texturing engineering guarantees the novelty and creativity of this study and achieved high piezoelectric properties and excellent temperature stability simultaneously.
In order to achieve excellent temperature stability of piezoelectric properties, the ratios for stacked multilayer composite ceramics were selected to be 1:3, 1:4, 1:5 and 1:6. The room temperature d33 and Suni values, and their temperature stability of multilayer composite ceramics with stacked ratios of 1:3, 1:4, 1:5 and 1:6 are presented in Figure 2. It can be seen the optimal piezoelectric properties (d33~467±16 pC/N, Suni~0.17%) at room temperature and excellent temperature stability of piezoelectric properties were achieved in the 1:6 multilayer composite textured ceramics. Therefore, the ratio for stacked multilayer composite ceramics was determined to be 1:6. In addition, we can deduce that if the proportion of KNN-T3 continues to increase (e.g., 1:7, 1:8××××××), the role of KNN-T1 will decrease, so it is difficult to realize the complementary effect and resulting excellent temperature stability in the multilayer composite ceramics.

Figure 2. Temperature dependence of (a) d33(T)/d33(RT) and (b) Suni(T)/Suni(RT) values of the multilayer composite ceramics with stacked ratios of 1:3, 1:4, 1:5 and 1:6. (c) Comparation of room temperature d33 and Suni for the multilayer composite ceramics with stacked ratios of 1:3, 1:4, 1:5 and 1:6.
Comments 3: Authors claimed that “(i) KNN-T1 ceramics exhibit three distinct dielectric anomaly peaks away from room temperature, corresponding to rhombohedral-orthorhombic (R-O) phase transition.” However, distinct evidence of R-O phase transition was not detected in Figure 2. Please make sure this part. What is meaning of “BKH” in 1st paragraph of page 4? Furthermore, authors claimed “1:6 multilayer composite ceramics possess a phase structure with coexisting R-O-T phases (i.e., polymorphic phase boundary, PPB).” However, the provided result is insufficient for supporting author's explanation.
Response: Thank you sincerely for your valuable comments and insightful suggestions. To accurately determine the phase transition through temperature-dependent dielectric constant (εr-T) curves of KNN-T1 ceramics (Figure 3a), a magnified curves in the temperature range of -60 oC—-20 oC is provided in Figure 3(b). In Figure 3(b), the dashed line serves as a reference line, and it can be clearly observed that there is a broad dielectric peak in the range of -60 °C to -20 °C, corresponding to the R-O phase transition of the KNN-T1 ceramics.
We are very sorry for our negligence of giving the meaning of “BKH” in 1st paragraph of page 4. In this work “BKH” refers to (Bi0.5K0.5)HfO3, which is added to the revised manuscript.
It is worth noting that the phase structure of 1:6 multilayer composite ceramics was determined by the temperature dependence of dielectric constant (εr-T) curves and XRD pattern as well as Rietveld refinement of XRD pattern. One can see from εr-T curve the R-O phase transition and O-T phase transition merged into a R-O-T phase transition at approximately room temperature. Furthermore, Rietveld refinement of XRD pattern is accepted to be an effective method to determine the phase structure of ceramics. Therefore, the Rietveld refinement of XRD pattern for the 1:6 multilayer composite ceramic was performed and the result is given in Figure 3(c). The reliability factor of weighted patterns (Rwp), the reliability factor of patterns (Rp) and the goodness-of-fit indicator (χ2) are 3.81%, 2.95% and 1.32, respectively, indicating that the structural model is valid and the refinement results is reliable. As a result, the phase structure of 1:6 multilayer composite ceramic was determined to be the coexistence of R, O and T phases.

Figure 3. (a) Temperature-dependent dielectric constant (ɛr-T) curves measured at 100 kHz. (b) The enlarged image of the ɛr-T curve for KNN-T1 ceramics in the temperature range of -60 oC to -20 oC. (c) Rietveld refinement of the XRD pattern of 1:6 multilayer composite ceramic.
Comments 4: It seems that the provided results for temperature stability (both of d33 and strain properties) of 1:6 multilayer composite textured ceramics are difficult to find the advantage in comparison with a single KNN-T3.
Response: Thank you for your valuable comment. The single KNN-T3 ceramic exhibits large variations of approximately 64.6% and 10.6% in d33 and unipolar strain, respectively, while low fluctuations of only 9.1% and 2.9% are observed in the 1:6 multilayer composite ceramics over the temperature range of 25 oC-180 oC. Therefore, a significant advantage was observed in the 1:6 multilayer composite ceramics in comparison with a single KNN-T3.
Once again, thank you very much for your valuable and constructive comments.
Round 2
Reviewer 1 Report
Comments and Suggestions for Authors
The authors should include their comments on the melting temperature and SEM images (responses 1 & 2) in the manuscript.
Author Response
Comments 1: The authors should include their comments on the melting temperature and SEM images (responses 1 & 2) in the manuscript.
Response 1: Thank you very much for your constructive and valuable suggestion. We are very sorry for our negligence of adding the comments on the melting temperature and SEM images to the manuscript. It is well-known that the melting temperature of KNN is at approximately 1140 oC – 1150 oC. However, the reasons why a high temperature (1180 oC) was designed in the first step during the two-step sintering process have been added to the revised manuscript. (Page 9, line 301-304) In addition, the reasons why a specific grain morphology was not observed in the cross-section of KNN have been added to the revised manuscript. (Page 4, line154-159)
Once again, thank you very much for your kind work and constructive comments. On behalf of my co-authors, we would like to express our great appreciation to you for recognition and approval of our paper.
Reviewer 2 Report
Comments and Suggestions for Authors
It seems that the revised manuscript is well strengthened and improved. Therefore, current form is ready to publish in Molecules
Author Response
Comments 1: It seems that the revised manuscript is well strengthened and improved. Therefore, current form is ready to publish in Molecules.
Response 1: Thank you very much for your kind work and constructive comments. On behalf of my co-authors, we would like to express our great appreciation to you for recognition and approval of our paper.